# Comprehensive Profiling of Primary and Metastatic ccRCC Reveals a High Homology of the Metastases to a Subregion of the Primary Tumour

**DOI:** 10.3390/cancers11060812

**Published:** 2019-06-12

**Authors:** Paranita Ferronika, Joost Hof, Gursah Kats-Ugurlu, Rolf H. Sijmons, Martijn M. Terpstra, Kim de Lange, Annemarie Leliveld-Kors, Helga Westers, Klaas Kok

**Affiliations:** 1Department of Genetics, University of Groningen, University Medical Center Groningen, HPC CB50, 9700 RB Groningen, The Netherlands; p.ferronika@umcg.nl (P.F.); j.hof01@umcg.nl (J.H.); r.h.sijmons@umcg.nl (R.H.S.); m.m.terpstra.cluster@gmail.com (M.M.T.); k.de.lange@umcg.nl (K.d.L.); h.westers@umcg.nl (H.W.); 2Department of Pathology, Faculty of Medicine, Universitas Gadjah Mada, Public Health, and Nursing, Sekip Utara, 55281 Yogyakarta, Indonesia; 3Department of Pathology and Medical Biology, University of Groningen, University Medical Center Groningen, HPC EA10, 9700 RB Groningen, The Netherlands; g.kats-ugurlu@umcg.nl; 4Department of Urology, University of Groningen, University Medical Center Groningen, HPC CB60, 9700 RB Groningen, The Netherlands; a.m.leliveld@umcg.nl

**Keywords:** intratumour heterogeneity, metastatic ccRCC, copy number alteration, mutation, gene expression

## Abstract

While intratumour genetic heterogeneity of primary clear cell renal cell carcinoma (ccRCC) is well characterized, the genomic profiles of metastatic ccRCCs are seldom studied. We profiled the genomes and transcriptomes of a primary tumour and matched metastases to better understand the evolutionary processes that lead to metastasis. In one ccRCC patient, four regions of the primary tumour, one region of the thrombus in the inferior vena cava, and four lung metastases (including one taken after pegylated (PEG)-interferon therapy) were analysed separately. Each sample was analysed for copy number alterations and somatic mutations by whole exome sequencing. We also evaluated gene expression profiles for this patient and 15 primary tumour and 15 metastasis samples from four additional patients. Copy number profiles of the index patient showed two distinct subgroups: one consisted of three primary tumours with relatively minor copy number changes, the other of a primary tumour, the thrombus, and the lung metastases, all with a similar copy number pattern and tetraploid-like characteristics. Somatic mutation profiles indicated parallel clonal evolution with similar numbers of private mutations in each primary tumour and metastatic sample. Expression profiling of the five patients revealed significantly changed expression levels of 57 genes between primary tumours and metastases, with enrichment in the extracellular matrix cluster. The copy number profiles suggest a punctuated evolution from a subregion of the primary tumour. This process, which differentiated the metastases from the primary tumours, most likely occurred rapidly, possibly even before metastasis formation. The evolutionary patterns we deduced from the genomic alterations were also reflected in the gene expression profiles.

## 1. Introduction

Kidney cancer is a usually lethal urologic malignancy with an annual mortality rate of 50% and an annual incidence of 337,000 new cases worldwide [1]. About 30% of all kidney cancer patients have metastases at the time of diagnosis, and another 30–40% will develop metastases at a later stage [2]. Clear cell renal cell carcinoma (ccRCC) is the most common type of kidney cancer in adults and metastasizes hematogenously to lungs, bone and liver [3].

Although somatic mutation and RNA expression profiles of primary ccRCC have been extensively described in the Cancer Genome Atlas project [4], the genomic profiles of metastatic ccRCC have not been frequently studied in the context of their primary tumours. In the few reported cases, intra primary tumour and metastasis heterogeneity was identified based on differences in copy number aberrations (CNAs) [5], single nucleotide variants (SNVs) [5,6] and RNA expression patterns [5,7].

Understanding how molecular (genomic) alterations accumulate during tumour evolution is important to gain insight into the development of metastasis and might provide clues to more optimal treatment strategies. As the majority of metastases are thought to establish through hematogenous dissemination [8], studying venous tumour thrombus samples may reveal the mutations of at least some of the cancer cells on their road to distant metastasis [9].

In this study, we extensively analysed samples from a unique ccRCC patient for whom nine samples were available taken from multiple regions of a primary ccRCC tumour, tumour thrombus reaching inferior vena cava, and four pulmonary metastases from different sites in the lung. We analysed the pattern of CNAs, SNVs, and gene expression in these samples to interrogate the process of tumour evolution in this patient. To explore the differences in gene expression among primary ccRCCs and metastases, we then analysed samples from four additional patients.

## 2. Results

### 2.1. Copy Number Profiles of Primary and Metastatic ccRCC

Array comparative genomic hybridization (CGH)-based copy number profiles were generated for three primary tumour samples (Pr1, Pr3, and Pr4), the inferior vena cava tumour thrombus (VT), and four lung metastases (M1–M4) of the index patient (Figure 1). All samples had a gain of 5q and a loss of chromosome 3, a small segment of 2q and 10q. In addition to these consistent aberrations, several CNAs were present in either a single sample or a subset of the primary tumour samples. We also observed intermediate copy number states at varying positions in all primary tumour samples, which was indicative of intra-sample heterogeneity in copy number levels. Clear examples of this were a gain of chromosome 2 in Pr1 and loss of 1p and chromosome 4 in Pr3. The most striking copy number differences in the primary tumour samples were observed between Pr1 and Pr3, on the one hand, and Pr4, on the other. Two different samples of primary tumour, Pr1 and Pr3 showed very similar CNA patterns. In contrast, Pr4 showed more extensive CNAs that closely resembled the patterns of the VT and the lung metastases, which shared the loss of 1p, 9 and 13q and the gain of chromosome 7, 12p and 20q as their common feature. The lung metastasis that developed after treatment with pegylated (PEG)-interferon, M4, was distinct from other metastases by several copy number alterations, most prominently the loss of 11q. 

A phylogenetic tree based on the CNA data (Figure 2) shows a clear separation, with two primary tumour samples (Pr1 and Pr3) on one branch and the third primary tumour sample (Pr4), VT, and the metastases samples on the other branch. In agreement with the CNA pattern, Pr4 was most closely related to M2, whereas the VT was almost identical to M3. Even though we noticed variations in tumour grade among primary tumours and metastases, we were unable to prove a relationship of regional tumour grade to specific copy number events.

We generated B-allele frequency (BAF) plots for all samples, including Pr2, for which we had no array CGH data. The BAF plot of Pr2 resembled that of Pr1 and Pr3, indicating a similar CNA pattern (Appendix A). We then used the BAF plots to determine absolute chromosomal copy numbers, and thus, the ploidy of the tumour cells. All metastases showed the lowest copy number states for chromosomes 6, 9, and 13 in the array CGH profiles. The BAF data, however, indicated an even number of copies for chromosome 13 in Pr4, VT, and the lung metastases (Appendix A). This suggests that the tumour cells of Pr4, VT, and the four lung metastases contained two copies of chromosome 13. The other genomic segments with a similar copy number state in the array CGH plots, including chromosome 3 in all samples except M4, also represented a copy number state of 2. The BAF plots also indicated that all copies of chromosome 3 in Pr4, VT, and the lung metastases originated from the same parental allele. The next level in the CNA plots in Pr4, VT, and the metastases should refer to chromosomal segments for which three copies were present. Indeed, the BAF plots of the germline variants for chromosomes 8, 14, and 15 indicated an odd number of copies in the tumour cells in Pr4, VT, and the lung metastases, consistent with the presence of three copies. Taken together, these data indicated that Pr1–Pr3 had a near diploid genome, while Pr4, VT, and the four lung metastases had a near tetraploid genome. 

### 2.2. Somatic Mutations Identified in Primary and Metastatic ccRCC 

Whole exome sequencing (WES) was conducted for all tumour samples of RC1 with matched normal kidney cortex used as control. The mean target coverage for all samples was 57×, with a coverage >10× for 90% of the target region (Appendix A). 

A total of 146 non-synonymous somatic mutations identified in 138 genes were defined as a major clone mutation in at least one tumour sample, adding up to 390 events of major clone mutations in nine tumour samples (Figure 3). In an additional 104 instances, these mutations were classified as minor clone mutations. Targeted sequencing to validate 27 randomly chosen somatic mutations (including both major and minor clone mutations) in nine tumour samples led to a total of 243 validation events, e.g., genomic positions where a mutation should either be present or absent in a specific sample (Appendix A). Coverage in the targeted sequencing data was sufficient for 207 events. Almost all of the variants, 203 out of 207 (98%), could be validated. The number of major and minor clone mutations that could be validated by targeted sequencing was similar: 127 out of 129 for major clone mutations and seven out of eight for minor clone mutations.

Eighteen of the 146 non-synonymous somatic mutations (12.3%) were shared by all tumour samples. Among these were mutations in two well-known ccRCC cancer driver genes (*VHL* and *PBRM1*) and in *EPHA4*, a lung adenocarcinoma driver gene [10]. Fifty-one mutations (35%) were present in a subset of the primary tumours but not detected in the VT nor in the lung metastases. This included a mutation in a known ccRCC cancer driver gene, *ARID1A*. Twenty mutations were unique for the metastases. We did not observe a clear increase in mutational load consistent with the evolutionary path reflected by CNA patterns depicted in Figure 2. Two primary tumour samples, Pr1 and Pr3 clearly separated in one branch of the phylogenetic tree (Figure 2) and had 18 and 10 unique mutations, respectively. For all other samples, the number of unique mutations fell within the range of 3–6. In Pr4, we identified 10 mutations that were not present in the other primary tumour regions, of which three (in *BLOC1S4*, *WDFY4*, and *CUBN*) were shared with VT and all lung metastases.

### 2.3. Gene Expression Profiling of Primary and Metastatic ccRCC Samples

We carried out a gene expression analysis to further explore the relationship between the primary tumour and the metastases in the index patient (Appendix A). Unsupervised hierarchical clustering based on the 500 most variably expressed genes showed a clear distinction between three of the primary tumour samples (Pr1, Pr2, and Pr3) and the remaining samples, including Pr4, VT, and the metastases (Figure 4). The four lung metastases clustered separately from VT and Pr4.

To evaluate the performance of gene expression data in visualizing the relationship of primary tumour segments to the metastases of the same patient, we generated gene expression profiles for multiple tumour samples of four additional ccRCC patients (Table 1). 

Unsupervised hierarchical clustering of the most variably expressed genes for each patient indicated that the most distinct separation was between the primary tumour samples and the metastatic samples (Appendix A). Within the primary and metastatic samples, the dendrogram branches tended to be further separated based on their regional grades and metastatic sites. In patient RC3, one primary tumour sample clustered together with the metastases, suggesting its role as the tumour region responsible for the metastasis, which was similar to what we observed in our index patient.

### 2.4. Differentially Expressed Genes in Primary Tumours versus Metastatic Tumours

We next identified genes consistently differentially expressed between primary tumours and metastases of all five patients. As the VT sample of the index patient (RC1) could not be clearly classified as a primary tumour or a metastasis, this sample was not included. The analysis identified 57 genes with an absolute fold change > 3 (*p* < 0.01), 32 of which were upregulated and 25 that were downregulated in metastases versus the primary tumours (Appendix A). The Database for Annotation, Visualization and Integrated Discovery (DAVID) annotation analysis [11] of these genes showed enrichment for blood micro-particle group (*p* < 0.009) (*CP*, *FGA*, *FGB*, *SERPINA3*, and *ALB*) and extracellular matrix organization (*p* < 0.03) (*ACAN*, *COL11A1*, *DCN*, *FGA*, *FGB*, *LAMA2*, and *LOX*). All differentially expressed genes that clustered in the blood micro-particle and extracellular matrix organization groups were upregulated in metastases, except for *ALB* and *ACAN*. Although not indicated as an enriched pathway, seven genes (*COL11A1*, *DCN*, *FGA*, *FOSB*, *SPOCK1*, *AQP9*, and *PTGDS*) in this list are related to the epithelial–mesenchymal transition pathway [12,13,14,15,16,17,18].

## 3. Discussion

Application of high-throughput sequencing to multiregional sampled ccRCC has highlighted a marked degree of intratumoural genomic heterogeneity [6,9], but cases with analysis of multiple primary tumour subregions and multiple metastases are still relatively rare. We present a metastatic ccRCC patient for whom multiple samples of the primary tumour, inferior vena cava, and lung metastases were available. Tumour grade heterogeneity is seen both in the primary tumour sections and metastases by haematoxylin and eosin staining. However, we did not observe any significant morphological feature that differentiated metastasis from primary tumour sections. In contrast, through multiregional sampling and extensive genomic analysis, we identified the somatic alterations underlying early and late events in tumour development. We also identified one subclone within the primary tumour that closely matched the profiles of the metastases and is, therefore, likely to be their origin. 

When we looked at the pattern of structural alterations to map tumour evolutionary events, we noted two distinct groups. One group, consisting of three primary tumour samples, had a near-diploid genome with few CNAs. The other group, consisting of primary tumour sample Pr4, the vena cava tumour thrombus (VT), and all lung metastases, had a complex, largely similar, copy number profile with tetraploid characteristics. We assumed that conversion of a near diploid tumour cell (Pr1–Pr3) to a near tetraploid state (Pr4) resulted in a tumour–cell lineage with metastatic potential in this patient.

Theoretically, the VT could have acted as a “distribution station” between the primary tumour subclone with metastatic potential and all other metastases, as has been observed by Turajlic et al [9]. Alternatively, and not surprisingly, the primary tumour subclone could have “sent out” multiple cancer cells in parallel, and some of these could have formed the venous thrombus while others metastasized to distant organs, bypassing that thrombus station. Although based on small copy number differences in the metastases group, M2 most closely resembles Pr4, whereas VT and M3 have virtually identical copy number patterns with their only difference being an intra-sample copy number heterogeneity for chr11 in M3. Our data, therefore, suggest that the latter scenario was the most likely one, and Pr4, rather than VT, was the origin of all the metastases we studied. In this scenario, a number of closely related subclones may all have emerged in the region of Pr4 that resulted in the slightly different copy number patterns that we observed in VT and M1–M3. The lung metastasis that developed after treatment of PEG-interferon, M4, featured additional structural alteration in 11q and more copy number levels than the rest of the metastases. However, all the CNAs that characterize the metastases were also present in M4. Thus, M4 seemed derived from the same tumour subclone as the previously developed metastases. The existence of such clones, including very small ones, in different types of cancer matching the distant metastasis profile, was demonstrated by single-cell sequencing experiments including some by our group [19].

Compared to the analysis of the copy number changes, the somatic mutation profiles of the tumours were less informative for reconstructing metastatic origin but did reveal early evolutionary steps. We did not observe a clear difference in the mutation load between Pr1, Pr2, and Pr3, on the one hand, and Pr4, VT, and the lung metastasis, on the other. Only 12% of the somatic mutations were shared between all primary tumour subregions and the metastases. These included mutations in the well-known ccRCC-specific cancer driver genes *VHL* and *PBRM1*, which were, therefore, most likely involved in the initiation of tumour development in this patient. A low number of shared mutations between primary tumours and their metastasis was also reported in a study where single primary tumour samples were compared with a single metastatic sample per patient [20]. We did not see a gradual increase of mutation load. Instead, both the primary tumour sections and metastases we analysed show a more-or-less equal number of unique mutations. The presence of these private mutations might reflect on-going parallel clonal evolution in each of the primary tumour sections and in metastasis regions [21]. It probably reflects independent clonal evolution after the structural variations were established in this patient. The combined data suggest punctuated evolution of structural variations occurred within a short window of time, after which the mutational load of different parts of the primary tumour and all the individual metastasis further increased independently. 

We identified several somatic mutations restricted to the different metastatic sites in the lungs, but none of these mutations were shared between all lung metastasis sites. This, again, indicates that metastasis-to-metastasis dissemination of cancer cells, e.g., as seen in prostate cancer [22], did not occur in this patient. Instead, all metastases appeared to have originated individually from the metastasis-resembling region of the primary tumour. 

With respect to the mutations that may have facilitated the process of metastasis, we found it interesting that we observed a missense mutation of *ARHGAP12* and a frameshift deletion of *CENPN* in the VT and all lung metastatic samples. Gene *ARHGAP12* encodes a junctional complex protein that affects tumour cell adhesion, scattering, and migration driven by hepatocyte growth factor [23,24]. These processes regulate invasive growth, and when aberrantly expressed in cancer cells, leads to cancer invasion and metastasis [25]. Downregulated *ARHGAP12* has been related to the increased invasive growth of human cancer cell lines from lung epithelial cells, prostate, thyroid, and breast [24]. As it is well known, *CENPN* is important in kinetochore assembly prior to mitosis [26]. Although no literature specifically mentions the involvement of these mutated genes in ccRCC, we speculate that these mutations contributed to metastasis in the index patient. These mutations may already have been present in the primary tumour, but if so, the number of tumour cells in our sample was too small for these mutations to be discovered. Few mutations were only present in area Pr4 of the primary tumour and were shared with VT and all lung metastases, including one missense mutation of *CUBN*. Being expressed in several normal epithelial cell types, including those in the kidney, *CUBN* encodes a receptor for intrinsic factor-vitamin B12 complexes [27]. Low *CUBN* expression has been found by others in venous tumour thrombus and lung metastases, as compared to primary tumours of ccRCC [28]. The same study found low *CUBN* expression was associated with poor overall and cancer-specific survival in ccRCC patients. In our patient’s tumours, we did not observe low *CUBN* expression. In fact, none of the genes that showed differential expression between primary tumour and metastases carried a mutation in any of the tumour samples. Conversely, the mutations that occurred in these tumours, apparently did not influence mRNA levels of the mutated genes.

To characterize tumour subclones in primary tumours and metastases based on gene expression, we analysed mRNA profiles of the index patient and four additional ccRCC patients. Through gene expression profiling, we were able to confirm the putative metastatic seeding subclone in the index patient. In the additional patients, we succeeded in one case (RC3). Our inability to identify the putative metastasis-seeding subclone in the other three patients was most likely due to the incomplete sampling and did not exclude the presence of such a subclone in the primary tumour. 

Among the 57 differentially expressed genes in the metastatic versus primary tumour samples, six extracellular matrix genes were upregulated, and one was downregulated in the metastases. We also observed upregulation of seven genes related to the epithelial–mesenchymal transition in the metastases, three of which are part of the extracellular matrix pathway as well. Together with intratumoural hypoxia, the extracellular matrix has been suggested to play a crucial role in metastasis development [29] through specific molecular pathways such as the epithelial–mesenchymal transition [30]. Interestingly, upregulation of extracellular matrix genes in the metastases of RCC patients has recently been described [31], which supports our observation. 

## 4. Materials and Methods 

### 4.1. Patient Materials

Formalin-fixed paraffin-embedded (FFPE) tissues were collected from four different regions of the primary tumour, from the venous tumour thrombus and from four metastatic lesions of one ccRCC patient—the index patient (Figure 5). With the exception of one lung metastasis (M4), all tumours were removed prior to PEG-interferon therapy. 

For four additional ccRCC patients, multiple regions of a primary ccRCC tumour and metastases were available (Table 1). Haematoxylin and eosin staining was used to grade each tumour sample histomorphologically according to the WHO/ISUP system 2012 [32,33]. The study was performed in accordance with the University Medical Center Groningen Medical Ethical Review board (project number 20190251, approved 4th January 2016) and Dutch ethical guidelines and laws, and complied with the regulations stated in the Declaration of Helsinki. The FFPE tissue section and DNA/RNA isolation are described in the Appendix A.

### 4.2. CNA Analysis

For all of the index patient’s tumour samples, array CGH was carried out using 500 ng genomic DNA from FFPE tumour samples using the Complete Genomic SureTag DNA Enzymatic Labelling Kit protocol and an OligoaCGH/ChIP-on-Chip Hybridization kit (Agilent, Santa Clara, CA, USA), according to the manufacturer’s instructions. Normal kidney cortex DNA isolated from FFPE material from the index patient was used as reference. Labelled DNA samples were hybridized on the Agilent Microarray, Custom HD-CGH, 4 × 180 K (Agilent, Santa Clara, CA, USA) following the manufacturer’s protocol. After scanning of the arrays, data were analysed with Nexus 7.5 software (BioDiscovery, El Segundo, CA, USA).

### 4.3. Whole Exome Sequencing

All tumour samples from the index patient were subjected to whole exome sequencing (WES) (Beijing Genomic Institute, China). Exome capturing and subsequent library preparation were conducted using 100 ng of DNA isolated from FFPE material using SureSelect-All exon V2™ (Agilent, Santa Clara, CA, USA). The final library was quantified using an Agilent 2100 Bioanalyzer (Agilent, Santa Clara, CA, USA). Paired-end sequencing with 100 bp reads was performed on the Illumina HiSEQ 4000™ (Illumina, San Diego, CA, USA). Sequencing data were processed using our in-house bioinformatics pipeline (https://github.com/mmterpstra/molgenis-c5-TumorNormal/tree/459417cc9553fae8c3040953970938860dafdfea), as described previously. The GATK HaplotypeCaller (downloaded from https://software.broadinstitute.org/gatk/) was used as variant caller [34]. Variant filtering and somatic mutation identification are described in the Appendix A. All sequencing data are available in the European Nucleotide Archive (ENA) repository (accession number PRJEB32862). To evaluate the CNAs identified by array CGH in more detail, B-allele frequency plots based solely on germline variants were generated for all tumour samples.

### 4.4. RNA Sequencing

The RNA libraries were prepared for tumour samples from the index patient and four additional patients using the QuantSeq 3’ mRNA-Seq library prep kit (Lexogen, Vienna, Austria), according to the protocol for degraded (FFPE) RNA. The enriched libraries were sequenced on the Illumina NextSeq 500 System (Illumina, San Diego, CA, USA). Processing of raw sequence reads and subsequent gene expression analysis are described in the Appendix A.

## 5. Conclusions

Copy number alteration and somatic mutation profiling from multi-region sampling of the primary tumour and metastases facilitates the identification of somatic alterations that underlie early events in tumour evolution and subsequent events in metastatic development. Gene expression profiling can reveal additional alterations at the transcriptome level. Together, these techniques may help to further identify the genomic profiles in primary renal cell cancer and their metastases, with the ultimate goal of finding patterns that can improve diagnostics and guide clinical management of this severe condition. 

## Figures and Tables

**Figure 1 cancers-11-00812-f001:**
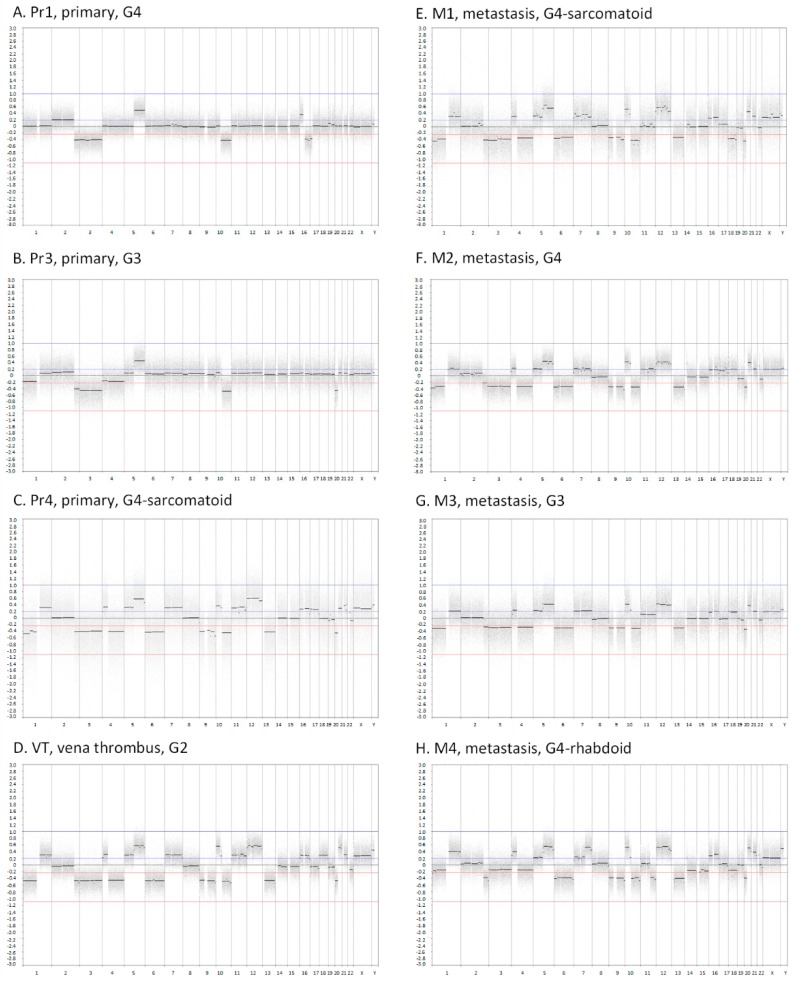
Array comparative genomic hybridization (CGH) plots of eight tumour samples from ccRCC patient 1 (RC1). The *x*-axes show the genomic position starting from 1pter until Xqter. The *y*-axes indicate the log2 intensity ratio between tumour and reference. Abbreviations: Pr1, primary tumour 1; Pr3, primary tumour 3; Pr4, primary tumour 4; VT, inferior vena cava tumour thrombus; M1, metastasis 1; M2, metastasis 2; M3, metastasis 3; M4, metastasis 4; G1, tumour grade 1; G2, tumour grade 2; G3, tumour grade 3; G4, tumour grade 4.

**Figure 2 cancers-11-00812-f002:**
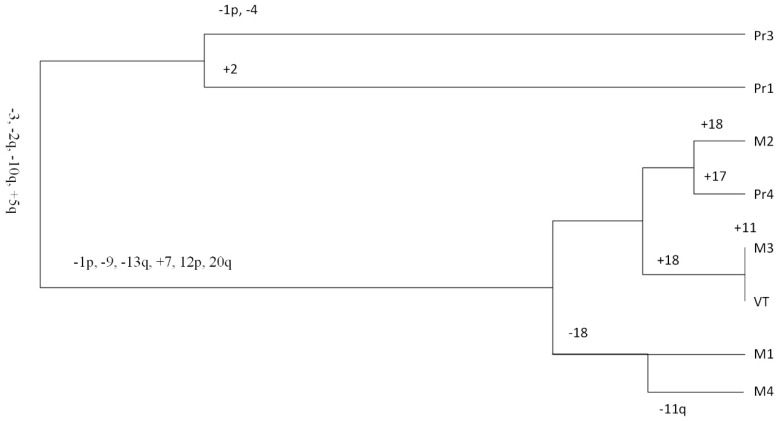
Phylogenetic tree based on copy number alteration (CNA) characteristics from different tumour samples of ccRCC patient 1 (RC1). Abbreviations: Pr1, primary tumour 1; Pr3, primary tumour 3; Pr4, primary tumour 4; VT, inferior vena cava tumour thrombus; M1, metastasis 1; M2, metastasis 2; M3, metastasis 3; M4, metastasis 4.

**Figure 3 cancers-11-00812-f003:**
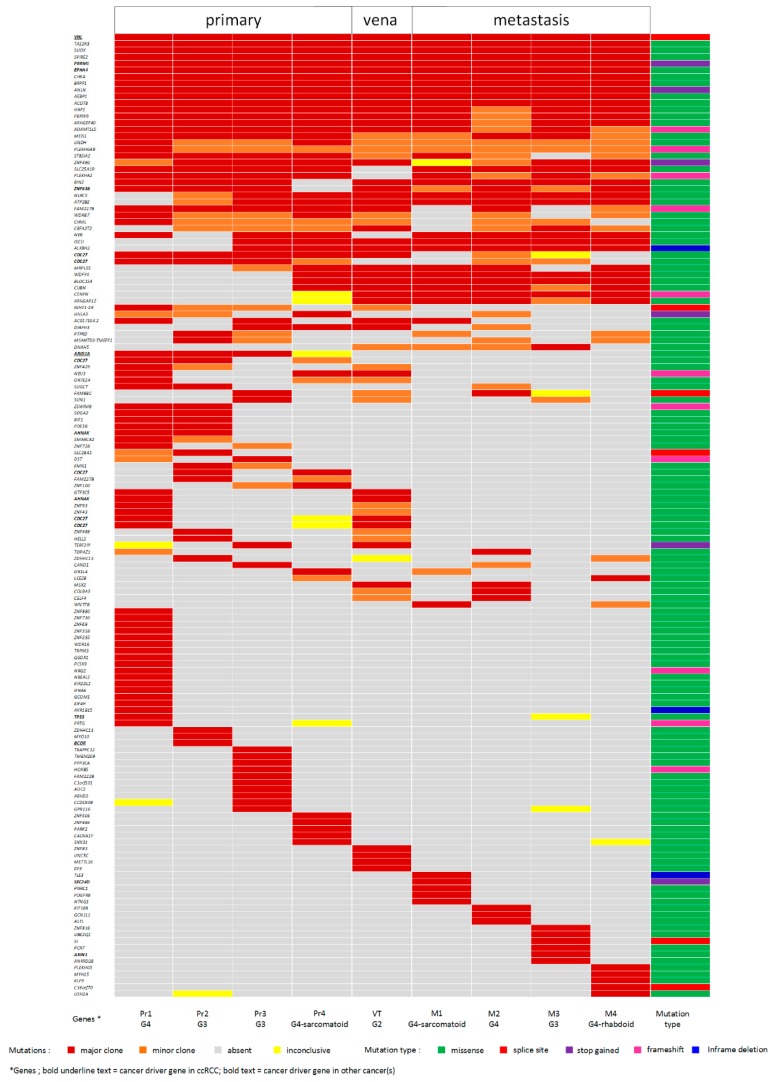
Somatic mutations identified in multiple regions of the primary tumour and metastases detected by whole exome sequencing (WES). The somatic mutations were classified as major or minor clonal as described in the Material and Methods section. Classification of mutations is indicated by the colours in the legend. The somatic mutations encompass 138 genes, including 11 cancer driver genes, as highlighted at the bottom of the figure. Abbreviations: Pr1, primary tumour 1; Pr2, primary tumour 2; Pr3, primary tumour 3; Pr4, primary tumour 4; VT, inferior vena cava tumour thrombus; M1, metastasis 1; M2, metastasis 2; M3, metastasis 3; M4, metastasis 4; G2, tumour grade 2; G3, tumour grade 3; G4, tumour grade 4.

**Figure 4 cancers-11-00812-f004:**
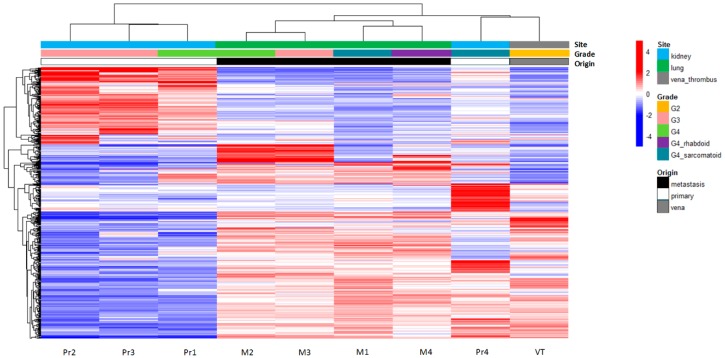
Unsupervised hierarchical clustering of expression profiles generated for ccRCC patient 1 (RC1). Clustering was based on the 500 genes with the highest variance in gene expression across all samples. The respective tumour samples are indicated at the bottom of the graph. For each gene, the mean value across the samples was defined. Expression levels higher than the mean of all samples are indicated in red. Expression levels lower than the mean are indicated in blue. Site of origin and the tumour grade are indicated by the colours at the top of the figure. The overall similarity between tumour samples is depicted by the dendrogram at the top of the figure and is based on the measurement of the Euclidian distance between tumour samples in expressing genes. Abbreviations: Pr1, primary tumour 1; Pr2, primary tumour 2; Pr3, primary tumour 3; Pr4, primary tumour 4; VT, inferior vena cava tumour thrombus; M1, metastasis 1; M2, metastasis 2; M3, metastasis 3; M4, metastasis 4; G2, tumour grade 2; G3, tumour grade 3; G4, tumour grade 4.

**Figure 5 cancers-11-00812-f005:**
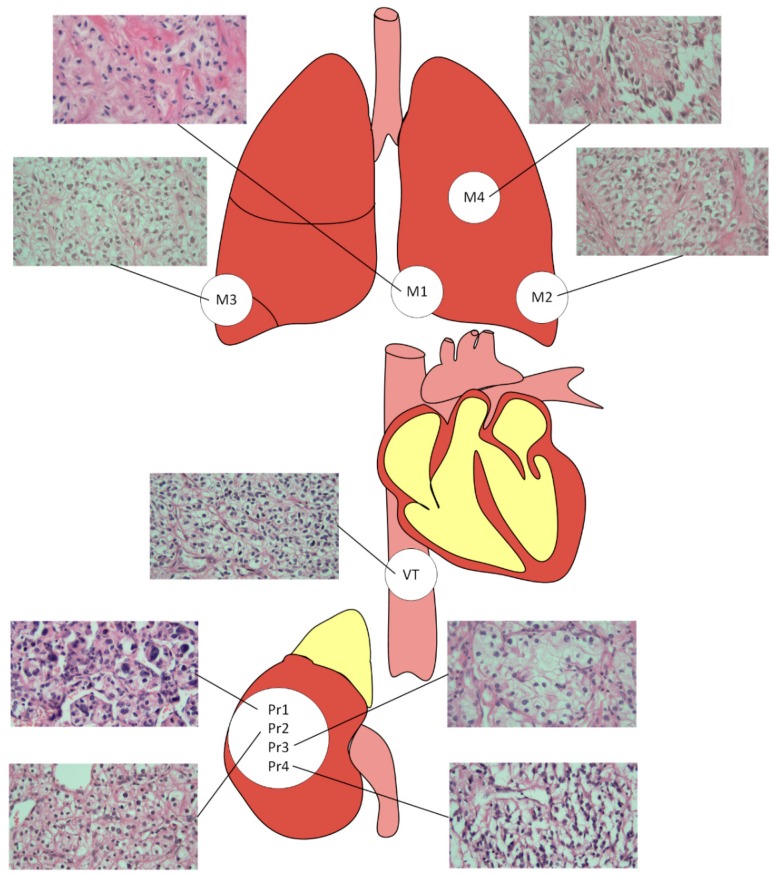
Origin of tumour samples. The primary tumour samples were grade 4 (Pr1), grade 3 (Pr2), grade 4 with sarcomatoid differentiation (Pr4), and grade 3 (Pr3). The venous tumour thrombus of 10 cm length extending to the inferior vena cava was grade 2 (VT). The metastasis in the lingula of the left lung was grade 4 with sarcomatoid differentiation (M1). The metastasis in the dorsal apex, the lower lobe of the left lung, was grade 4 (M2). The metastasis in the lateral basal, the lower lobe of the right lung, had a tumour grade 3 (M3). The metastasis in the upper lobe of the left lung was grade 4 with rhabdoid differentiation and was obtained by lobectomy after PEG-interferon treatment (M4). The haematoxylin and eosin -stained images were made based on 400× magnification.

**Table 1 cancers-11-00812-t001:** Tumour sample characteristics.

Patient	Tumour Origin	Tumour Grade (WHO/ISUP 2012)						Metastatic Site	Method Applied
		G1	G2	G3	G4-epitheloid	G4-sarcomatoid	G4-rhabdoid		
RC1 (pT3bN0M1) Index patient	4 primary samples	-	-	2	1	1	-		WES, Array CGH, RNAseq
	1 vena thrombus	-	1	-	-	-	-	inferior vena cava	
	3 metastasis pre-treatment	-	-	1	1	1	-	lung	
	1 metastasis post-treatment	-	-	-	-	-	1	lung	
RC2 (Pt3NoM1)	4 primary samples	-	3	1	-	-	-		RNAseq
	5 metastasis samples	-	-	3	2	-	-	brain	
RC3 (pT1bN0M1)	4 primary samples	-	-	2	2	-	-		RNAseq
	4 metastasis samples	-	-	-	4	-	-	brain, omentum	
RC4 (pT1N0M1)	4 primary samples	-	2	2	-	-	-		RNAseq
	3 metastasis samples	-	2	-	1	-	-	costae	
RC5 (pT3N0M1)	3 primary samples	-	3	-	-	-	-		RNAseq
	3 metastasis samples	-	1	2	-	-	-	lung	

Abbreviations: pT, pathological tumour size; N, lymph node involvement; M, metastasis; G, tumour grade (World Health Organization/International Society Urological Pathology system, 2012); WES, whole exome sequencing; Array CGH, array comparative genomic hybridization; RNAseq, RNA sequencing.

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
