# Peer review of "Comprehensive Profiling of Primary and Metastatic ccRCC Reveals a High Homology of the Metastases to a Subregion of the Primary Tumour"

_cancers, 2019, doi:10.3390/cancers11060812_

Round 1
Reviewer 1 Report
Ferronika et al. investigated the genomes and transcriptomes of primary clear cell renal cell carcinoma (ccRCC) and those of matched metastases to compare their genomic profiles. The copy number profiles suggest that a evolution from a subgroup of primary ccRCC, and that it probably occurred even before metastasis formation. Criticisms: This study is interesting and may play a large role in the research of ccRCC. The morphological heterogeneity of ccRCC is also well known, and it is common to form high-grade cancer nodule within low-grade ones. I think that more readers will be able to understand the significance of this paper more easily by adding HE-stained figures of primary and metastatic tumors to compare genetic findings and morphological features.
Author Response
Reply to the Reviewer 1:
We thank the reviewer for the invested time to review our manuscript and for suggestions to improve our paper. We agree that tumoural heterogeneity is of interest both at a clinical/pathological and genomic level, and it is worthwhile to try to find its role in the clinical management of ccRCC patients.
As suggested by the Reviewer 1, we now have included pictures of the HE-staining of all tumour areas from the index patient (RC-1) in Figure 5. Contrary to the genomic findings, morphological findings did not clearly differentiate between primary tumours and metastases. We have added this remark to the discussion. At this point we decided not to include HE-staining pictures of the additional patients as it would in our opinion add little to the discussion. However, we could surely include this data as an additional supplementary file. The assembly of such a figure would, however, take a little more time to prepare.
One sentence was added to the legend of Fig 5:
l. 296-297: The H&E-stained images were made based on 400x magnification.
Reviewer 2 Report
Interesting assessment of primary and metastatic ccRCC.
Major Points:
- Why were you unable to perform copy number profiling on Pr2?
- Please clarify the layout/labeling of Table 1. Which RC# is the primary patient for this study (RC1) and which are the four additional cases (RC2-5)?
Minor Points:
- Why was IFN chosen as the treatment of choice? That is not a contemporary standard of care for metastatic ccRCC.
- Please comment on the applicability of your findings given the very small sample size.
Author Response
Reply to Reviewer 2:
We thank Reviewer 2 for taking time to review our manuscript. The Reviewer 2 asked why we did not carry out copy number profiling on Pr2. Our arrayCGH analyses were carried out in multiples of 8 samples. Analyzing a ninth sample would thus have been very costly. As the B-allele frequency plots of Pr1, Pr2, and Pr3 were very similar and the B-allele frequency plots of all samples correspond well with the arrayCGH analyses we decided not to carry out an arrayCGH analysis for Pr2.
The reviewer asked us to clarify the labelling of Table 1. We apologize for having been not clear enough in this table. Table 1 has been modified. RC1 has now been indicated as the index patient in Table 1.
The reviewer also asked why IFN was chosen as the treatment. The index patient was managed in our hospital from 2004 to 2007. In that time interferon was the first choice for metastatic RCC treatment in our hospital, in accordance with the European guidelines [1].
1. Ljungberg, B.; Hanbury, D. C.; Kuczyk, M. A.; Merseburger, A. S.; Mulders, P. F.; Patard, J. J.; Sinescu, I. C.; European Association of Urology Guideline Group for renal cell, c., Renal cell carcinoma guideline. Eur Urol 2007, 51 (6), 1502-10.
The reviewer asked us to comment on the very small sample size. Of course we have to agree with the reviewer that the sample size is small. However, the number of primary ccRCC + metastatic lesions that are studied and reported with this level of detail is still very limited. Especially cases with multiple metastases are rarely available for study in clinical practice. Our study does provide a clear insight in the scenario on how subsequent genomic alterations take a role in the metastatic development of a specific ccRCC patient. Although the clinical applicability of this study may be limited at this moment, it will give directions to future studies so that eventually we can link genomic data to treatment options for individual patients. Our study clearly demonstrates the importance of addressing intra-tumour heterogeneity, and shows how we can address this at the genomic and transcriptome levels. We think this has been addressed properly in the “Conclusions” section.